# Structural Disadvantage in Housing Opportunities

Nigel de Noronha 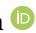

School of Social Sciences, University of Manchester, Manchester M13 9PL, UK; nigel.denoronha@manchester.ac.uk

**Abstract:** This paper discusses the ways that race and migration have shaped the housing opportunities people experience in England. It explores the historical development of policy and practice that has shaped racial inequalities in housing. It argues that the violence created by national and local state-supported housing policies has disproportionately affected racialised minorities, as has the slow violence generated by the neglect and stigmatisation of working-class housing. In turn, this has provided the justification for clearances and the remaking of space for those with the money to invest in the financialisation of land and housing through dispossession and denial of the right to safe, secure, and affordable housing. This analysis will be used as a basis to propose ways in which housing research can develop a coherent, critical perspective to race and migration and develop an alternative discourse to challenge the dominant market-driven, individualistic narratives. Adopting a critical approach allows researchers to move beyond the logic of housing policies at national and local levels to analyse and propose action to address persistent racial inequalities in housing.

**Keywords:** housing; inequality; race; racism; migration

## 1. Introduction

This paper develops a rationale and programme for critical research that addresses the ways that structural inequalities around race and migration history affect housing outcomes. It is intended to provide a provocation to housing scholars who have paid relatively little attention to race equality (Robinson et al. 2022). It is developed from my reflections on research into housing and race and provides ways in which these failings can be addressed. The lack of focus on race and migration within housing studies is particularly concerning given the response of public and civic organisations to Black Lives Matter. Whilst organisations have made commitments to address race equality within their own policies and practices, many outside these organisations have challenged institutions to do more and questioned the celebratory tones of some of the claims made about what is already being done. The challenge to the charity sector from Charity So White illustrates campaigning activities motivated by the Black Lives Matter movement.

> Our vision is of a charity sector that is taking the lead on tackling and rooting out racism. We want to see a shift in fundamental structures across the charity sector, where our sector, leaders and decision-makers reflect the communities that we work with. Unless we take serious and urgent action to tackle racism, social justice will not and cannot prevail. This will take investment and commitment and means leaders prioritising taking action and accountability, in order to bring about systemic change. Charity So White Vision Statement

Whilst this call to the charity sector is relevant to most social housing providers, the evidence of sustained approaches to addressing racism within the sector is limited. This failure is reflected in the disillusion amongst racialised minorities who have supported public service reform and contributed to the protests around Black Lives Matter in the UK.

Underpinning this article is the UN convention on social and economic rights, which the UK is a signatory to and which forms part of regular reports to them from the Equality and Human Rights Committee (UN 1976). As a signatory, the UK recognises the right of

everyone to adequate housing and to the continuous improvement of their living conditions. Research into housing is challenging given the different contexts and the way that housing policy changes over time in individual regimes. In terms of the public provisions of housing, the challenge is compounded in the UK's case by the delegation of housing policy to each country in the UK, and within England, the conflicting ways that housing policies are interpreted and implemented at local level. In the private sector, other actors such as landlords, builders, and estate agents also contribute to the complexity of housing research. As a result, the knowledge we generate through critical housing research is contingent on our understanding of the contexts we explore and the perspective we adopt when doing so. I believe it is important for us to be transparent about these views and the decisions we make about the evidence we collect and the ways we analyse and present our findings. Critical reading of housing policy demonstrates the way that "common-sense" facts are implicitly assumed and contribute to housing inequalities.

The second section develops a definition of racism that informs a critical appraisal of housing research. This critical approach enables researchers to move beyond the constraints of research that relies on evidence collected from those engaged with the provision of housing and recognises the importance of the lived experience of housing to critical housing research (Allen 2009). A summary of the methodologies used to gather the evidence presented to shape this critical approach is followed by three empirical chapters. The first empirical chapter explores the way that migration and race have been intertwined in historical and contemporary debates and have informed both policy development and political and media discourses; the second summarises some of the ways that this has shaped housing inequalities experienced by racialised minorities; and the last empirical section considers the failure of national housing policies to address these inequalities. The final section discusses what critical housing research that promotes equitable housing outcomes would look like.

## 2. Race and Migration

In this article, I use the term racialised minorities to reflect the ways that ethnic and other groups have been subjected to processes of being defined as racial others and the subsequent unequal treatment they have received as a result (Garner 2017; Solomos 2023). Despite the UK being one of the first signatories to the UN Convention of Racial Discrimination, national policy has been characterised by a failure to implement recommendations from commissioned reports and analyses, showing the ineffectiveness of government policy to sustainably address race equality over the last fifty years (Ashe 2021). In the social policy field, the lack of attention to race has been highlighted for many years (see, for example, Craig 2007, 2013; Craig et al. 2012). Within housing research, there has been limited critical engagement with the impacts of racism on the lived experiences of racialised minorities in England (Robinson et al. 2022).

The ways that policy regimes define racism is inconsistent, and in recent years, government policy has tended to equate it with individual behaviours (HM Gov 2021; CoDE 2021). This is predicated on the belief that individual acts of intentional racism are limited to a few bad apples within an organisation and that other discrimination is unintentional and will be resolved by training in unconscious bias and cultural awareness[1]. Race and racism are fluid concepts developed in different places and different times to racialise specific groups and justify unequal treatment (Hall 1997). In this paper, racism in relation to housing research in England is conceptualised as having four dimensions. Structural racism is created by the legal and policy framework that excludes people from access to support and services because of their racial characteristics. This includes those excluded because of their real or perceived citizenship status (Lukes et al. 2019) and those identified as undesirable by exclusionary discourses promoted by politicians and the media. Institutional racism reflects the way that organisations develop discriminatory policies and procedures in the treatment of those who access or try to access their services. Interpersonal racism is based on individual acts of racism and reflects the subjective experiences of those reporting it.

Intrapersonal racism is the internalised belief in inferiority based on personal characteristics and is likely to be reflected in distrust and disengagement with support and services. These experiences of racism are based on a lack of choice or opportunity, unfavourable treatment, or poor outcomes when trying to do the things that matter to people.

Robinson et al. (2022) found that "national policy statements in England do not recognise "race equality in housing as a priority concern" (p4), that "there is a lack of attention to ethnic inequalities in housing within contemporary research and analysis" (p4) and that "there is little evidence of regulators taking action to challenge poor performance or promote good practice in relation to equality of opportunity" (p4).

### 3. Methodology

Hansard provides access to transcripts of the political debates surrounding the development of legislation in Parliament. A search of substantive debates using the term race equality between 1950 and 1965 identified five substantial debates: one in 1958 on the need for immigration control; one in 1960 on immigrants from the West Indies; one in 1961 on immigration control, housing, and immigration; one in 1962 on the second reading of the *Commonwealth Immigration Act*; and the debate on the first *Race Relations Act* in 1965. A thematic analysis of the four hundred pages of debates showed that the main focus of the debates was on regulating levels of immigration of "coloured people" from the New Commonwealth. This political concern was subsequently translated to legislation targeted at this group. For comparative purposes, a search of substantive debates on immigration was conducted between 1900 and 1905 on immigration and housing. This search identified around one hundred and fifty entries. Using word search within these entries revealed the debate about the impact of migration on housing in London, echoing the later debate. The section on housing and racism draws on archival research on slum clearances in Moss Side in Manchester in the 1960s and 70s, analysis of the Evidence for Equality National Survey on experiences of racism when seeking housing (Finney et al. 2024), and analysis of 2021 census data on housing and race. The final section draws together an interpretation of successive housing policies and an analysis of the minimal impact they have had on race and racism in housing.

### 4. The Migrant Folk Devil

For more than a century, the debate about housing has been framed by concerns about race and migration, as shown in the extract from a parliamentary debate in 1902. The debate was part of the build-up to the passing of the *Aliens Act* in 1905, which was intended to stop the immigration of those fleeing pogroms in Eastern Europe and Russia.

> Not a day passes but English families are ruthlessly turned out to make room for the foreign invaders. Many of these have been occupying their houses for years. . . . Out they have to go to make room for Romanians, Russians, or Poles. Rents are raised 50% to 100% and a house which formerly contained a couple of families . . . is made to hold four or five families. (Hansard 1902)

> The British workman is thus squeezed out of his home, and what happens? The house is immediately taken by five, six, eight, or ten of these aliens, who herd together under conditions which are at once degrading and insanitary. I know it has been said by some people that this is a racial question . . . This is not a question of Jew or Gentile. We are speaking of foreign paupers and aliens as a whole. When I use the word aliens, I refer not only to Russians and Poles, but also to Austrians and Italians—of whom there is a large colony, chiefly ice cream vendors and organ grinders, in Hatton Garden—and to the French immigrants in the neighbourhood of Soho. (Hansard 1902)

Whilst the debate leading up to the passing of the *Commonwealth Immigrants Act* in 1962 reflected other perspectives, the same underlying racism is evident in some of the contributions.

"...it cannot be right further to overcrowd already overcrowded slums. That is an obvious and humane consideration. It has nothing to do with colour; whether the extra overcrowding affects whites, browns, pinks or blacks, makes no difference...

...in England and Wales—excluding London, and the great cities of Glasgow and Edinburgh—about 450,000 houses are unfit for human habitation. To replace those slum dwellings will present the building industry with a great problem, and we should not forget that those slum houses are at present occupied by white English people. This extra burden on the building industry is in addition to that presented by the houses that have to be built to meet the natural increase in population...

...most immigrants come to the United Kingdom because they are poor and are looking for a better way of life with higher standards. It is because they are poor that they are driven, to begin with, at any rate, to the lowest rented districts, the slum areas...

...no more immigrants should be allowed into this country unless we have proper accommodation and proper sanitation for them, especially in view of the danger to health". (Hansard 1961)

The subsequent failure of the Labour Party to deliver their promises to repeal the 1962 Act reflected the growing consensus that immigration, in particular coloured immigration, was a vote loser. This was demonstrated in the general election of 1964, where "the country swung to Labour by 3.5 per cent but Smethwick swung to the right by 7.2 per cent as Tory Peter Griffiths won the seat on the slogan 'If you want a nigger for a neighbour vote Labour'" (Fisher 2018). The political decisions to introduce border controls specifically targeted at migrants from the New Commonwealth and the introduction of legislation on race equality contributed to the growth of research into race relations. Race, Community and Conflict (Rex and Moore 1967) explored the housing conditions of racialised minorities in Sparkbrook in Birmingham. The study used ethnic classifications to explain social relations within the neighbourhood and derived a set of housing classes to explain the differential outcomes they experienced. This approach presaged a focus on ethnicity, often used as shorthand for a set of essential characteristics associated with an ethnic group. This approach contributed to the replacement of scientific racism by culturally based explanations of the difference in outcomes. This shift moved the explanation from experiences of racism to the different behaviours of ethnic groups.

There is also a lack of clarity about the concept of ethnicity. In different national contexts, the definitions of ethnicity vary or are not available. In the UK, there are differences in the categories defined by the statistical agencies for England and Wales, Scotland, and Northern Ireland. In England, the categories combine concepts of nation, continent, colour, and religion. This encourages researchers to focus on the differences between groups and to ignore the heterogeneity of experiences within each ethnic group. Within housing research, the focus on ethnicity often fails to recognise the centrality of racism to the experiences of racialised minorities, and as a result, many research outputs can tend to be largely descriptive.

Research into the housing experiences of New Commonwealth migrants in the 1950s and 1960s demonstrates the centrality of race and racism to their experiences, highlighting the exploitative, racially motivated behaviour of landlords. The role of race was central to the housing crisis in London in the late 1950s and early 1960s (Davis 2001). He explains how the term Rachmanism came to describe the behaviour of slum landlords in maximising the extraction of rent from slum properties in London. The combination of migration from the Caribbean leading to increasing demand together with the removal of rent control led to:

"Rents in unfurnished accommodation ... doubled [between] 1957 and 1963 in cases of unchanged occupancy and almost trebled in cases where a change of occupancy had accelerated decontrol". (Davis 2001, p. 71)

Migrants from Bangladesh struggled to find safe housing in East London and resorted to mass squatting in existing properties in areas where there was at least some protection for the local community from racism and organised fascist activity nearby (Begum 2023).

Migrants to Moss Side in Manchester came both from the New Commonwealth and other parts of England, where there were issues finding jobs or accommodation (Ward 1975; GPI 1975; Renwick and Shilliam 2022). Originally a middle-class suburb outside the boundaries of the city and home to High Commissions and hotels to accommodate the international textile traders visiting the city, the growth of the city and the attraction of suburban living led to the larger houses being converted to "house farms" or lodging houses in the 1920s and 1930s for the growing number of industrial workers (Ward 1975). Migrants from the New Commonwealth came to these larger houses first and, as slum clearance plans were developed, bought up the housing that became available (Ward 1975). The early experiences of housing for migrants from the New Commonwealth were shaped by the denial of access to council housing through a waiting period before an application would be accepted. In Birmingham, this period was five years. For many, access to council housing was in the new system-built estates, many of which had significant structural issues (AIURR 1969; Rex and Moore 1967).

## 5. Racism(/s) and Housing

The structural racism created by the policies to stop primary immigration from the New Commonwealth and the media discourses that accompanied it through the 1960s and 1970s provide a particular example of racialisation of groups within society. There have been successive campaigns that have racialised specific groups (Garner 2017). These include the criminalisation of black youth in the 1970s (Hall et al. 1978), the development of Islamophobia in the 1990s and onwards (Runnymede Trust 1997), and hostility to asylum seekers and Eastern Europeans (Lukes et al. 2019). These discourses influenced the types of tenants that landlords would rent to. For many migrants, this meant that access to housing was reliant on working through their own networks rather than dealing with the racism they experienced (Ward 1975; GPI 1975; Selvon 1956).

These analyses of the experiences of migrants from the New Commonwealth demonstrate the way that structural and institutional racism has disproportionately affected access to housing during the 1950s and 1960s. The lack of access to council housing led many to invest in their own properties in the inner cities of England. Many of these were properties with a short life, as they were in areas designated for slum clearance (Renwick and Shilliam 2022; Ward 1975). Where slum clearance was agreed, this led to the compulsory purchase of their homes, and given the lack of availability of property at similar prices, this led many to be rehoused in council housing, often inferior and problematic. In Moss Side, the Housing Action group provided collective resistance to slum clearance, producing a regular newsletter with a circulation of one and a half thousand. They provided information about the clearance process and lobbied the council, demanding resident involvement in the plans for clearance. The response to their actions led to hostility from the Chief Planning Officer and local councillors, as well as racist responses to the community from the local councillor.

> "it is the experts who should be asking for permission to participate in the lives of the general public". (Moss Side News)

> "Moss Side Peoples' Association and Housing Action Group were run by nice people. They have been taken over by "black power boys, parasites and stirrers"". (local councillor)

> "The council should help immigrants who want to go home by giving them a decent price for their houses so that they can pay their own fares". (local councillor) (AIURR 1969)

Resistance to slum clearance provides important insights into the historical experiences of racialised minorities and the ways in which they accumulated financial disadvantages

compared to those in more favourable housing situations. Understanding the history of these communities' struggles provides an important perspective on what happened and the attitudes that this generated amongst racialised minorities regarding the role of key players in the local housing markets.

The persistence of these inequalities is evident from more recent data. Census data show persistent ethnic inequalities based on households experiencing overcrowding, sharing a kitchen or bathroom, or lacking central heating. The 2021 census for England and Wales shows that Bangladeshis were five times more likely to experience housing deprivation than the white British. Housing deprivation is associated with age, migration history, household composition, and tenure (ONS 2023a, 2023b, 2023c, 2023d, 2023e, 2023f). The relative privileges in terms of housing experienced by white British are likely to be based on their perceived belonging to the local area, whilst the differences between ethnic groups may be based on the cumulative advantages they have accrued. There are also class dimensions to housing privilege which have been relatively well researched in critical housing literature (Gray 2018; Rose 2024; Harloe 1995).

Figure 1 shows the extent to which different ethnic groups have experienced racism when seeking housing. The extent of differences resonates with earlier studies that highlighted the experiences of New Commonwealth migrants. These inform the commonly held view that research simply reflects how little racial inequalities have changed over time.

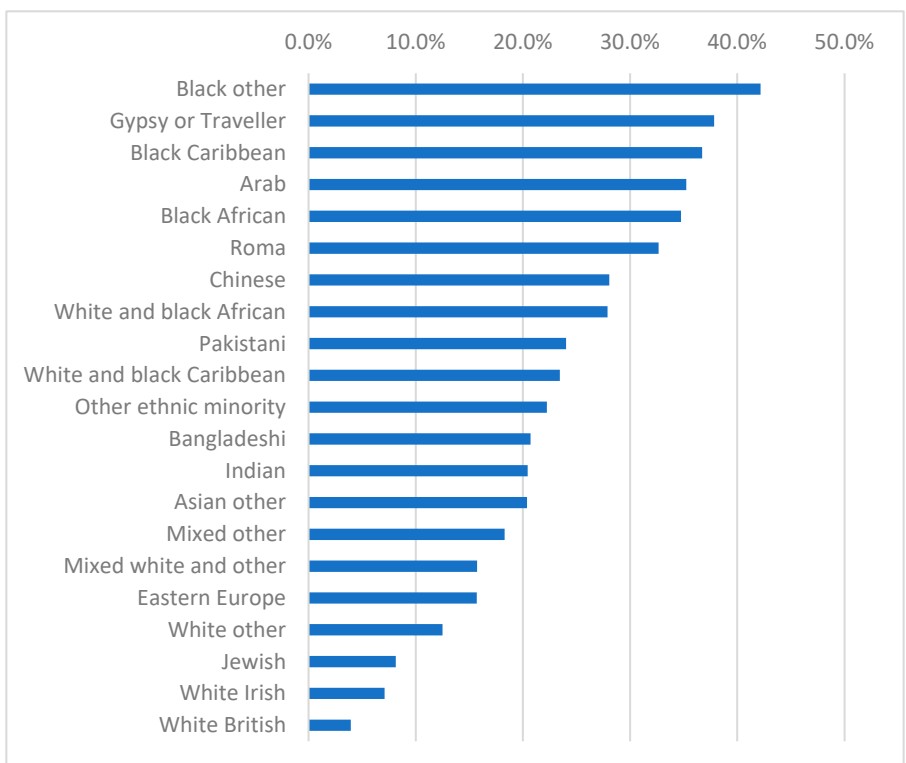

**Figure 1.** Experiences of racism when seeking housing by ethnic group (Finney et al. 2024).

The extent to which racialised minorities have experienced racism when seeking housing is stark, echoing reports from Shelter's recent survey of renters (BBC 2023). The extent to which these reflect historical patterns of racialisation deserves to be considered within local housing studies. The experiences of Gypsy and Traveller communities, the oldest ethnic minority in England, reflects government and public hostility that continues to deny adequate housing to meet their cultural needs and expectations (FFT 2023).

### 6. Housing Policy

The Milner Holland committee was set up in 1963 to investigate the housing crisis in London following the removal of rent controls in 1957. They focused on the issues of housing and race in Greater London. The committee found that racial discrimination in the housing market was endemic, with higher rents being charged to migrants (Davis 2001). This investigation led to the 1965 Rent Act, which established rent control under the fair rent mechanism. In 1964, the Housing Corporation was established to fund and regulate housing associations in England (Murie 2008). The report was commissioned by the Housing Corporation when it was closed in 2008, but does not engage with questions of race and housing (Murie 2008). The Housing Corporation introduced regulation of what had been a small part of the housing provision up until then and allocated capital funding to enable the development of more properties. In the 1980s, they funded BME-led housing associations to provide housing to meet the demand for housing for racialised minorities (Gulliver 2016). Whilst relatively small, this part of the sector continues to provide a model for combatting racism in the delivery of social housing.

Following the investment to address squalor after the Second World War, council housing, then the dominant provision of social housing, moved to building tower blocks and other system building designs to meet the demand for housing (Boughton 2018; Harloe 1995). Whilst the early provision was led by the commitment to build "homes fit for heroes" and was largely designed around garden estates, after the 1960s, it became an increasingly residual provision for those with no alternative options. The poor quality of some of this investment and the failure to adequately maintain it contributed to the stigmatisation of much of the newer housing stock (Johnstone and Mooney 2007; Hancock and Mooney 2013).

The introduction of the Right to Buy for council housing in 1980 reinvigorated the Conservative aim to build a property-owning democracy. As a result of the discounts available to existing tenants, most of the more desirable housing was transferred into private ownership (Harloe 1995). In 1988, rent for new tenancies in the private sector was deregulated, and the system is still operational today, with no rent control and limited security of tenure.

The common-sense assumption that ownership was the most desirable tenure dominated subsequent housing policies and was accompanied by a general neglect of the social housing stock, which was still mainly council housing. Meanwhile, the reduced regulation of the private rented sector is also likely to have contributed to a lack of investment by many landlords.

The election of New Labour in 1997 continued the dominant view that ownership was the most desirable tenure, and they left the Right to Buy legislation in place. Their priorities included a focus on social exclusion based on neighbourhoods, small geographical areas within local authorities. The focus on neighbourhood renewal, particularly in inner-city districts of major cities, was initially supported by piecemeal initiatives aimed at addressing employment and skills, education, crime, and housing. These were consolidated into a neighbourhood renewal programme underpinned by targets agreed upon by the government. In these neighbourhoods, housing was addressed by different programmes, including clearance, modernisation, development, and the creation of defensible space, applying design to deter crime (Reynald and Elffers 2009). The redevelopment of housing was required to meet a specified tenure mix to ensure the proportion of social housing did not exceed 40% (Bailey and Manzi 2008; Rowlands et al. 2006; Sautkina et al. 2012). Major investments were targeted at Housing Market Renewal areas, where market failure was diagnosed (Allen 2009; Bailey and Manzi 2008). This programme of area-based initiatives (ABIs) was designed to transform places, but paid relatively little attention to the people who lived there. Data on outcomes such as employment, education, and crime were collated to measure performance, but the characteristics of those who benefited from the investment were generally not collected. This could mean that the transformation of neighbourhoods was largely attributable to the attraction of new residents with different profiles. The longer-term plans for the physical regeneration of residential space included the assumptions that

a significant proportion of social housing was replaced with homes designed for ownership or private rent. The result was not only the displacement of some, but also the loss of space for the newly formed families with existing connections to the area. The new privately owned accommodation was characterised by many from the area as "not being for people like us".

Alongside this programme, there was investment in the remaining social housing stock to bring them up to a Decent Homes Standard[2]. As a pre-condition for accessing these funds, councils had to transfer their housing to specialist providers, existing or new Housing Associations, or council-owned housing management companies (Arms-Length Management Organisations or ALMOs). Whilst this transfer of ownership did not seem to have any direct consequences on the availability of social housing, it transferred control of local housing provision to a national regime. In hindsight, the consequences of this transfer were to enable national government to reshape the sector with initiatives that have included the requirement for them to set "affordable rents" set at 80% of market value (UN 2013), the requirement to sell more valuable properties, and tightening control over finances, limiting the ability to invest in new houses and encouraging the sector to become increasingly commercially oriented (UN 2013).

The financial crash in 2007/8 created significant issues for homeowners with unsustainable mortgages and for development plans within the private sector. Government bailouts of the banks in the UK meant that these did not lead to significant repossessions compared to other countries, but this support did not extend to sustaining the delivery of new construction projects.

More significantly, the crash led to the introduction of austerity policies alongside the removal of financial mechanisms that had allocated funding based on the level of need within areas. Councils requiring the most funding to support their residents had to deal with the largest financial cuts because of this redistribution. This led to reductions in the financial support available to tenants through the removal of the spare room subsidy (the bedroom tax), the reduction in the local housing allowance (LHA) to the 30th percentile of market rents within the local area, and the benefit cap limiting the overall benefits paid (UN 2013). The effect of the reduction in the LHA has made many places unaffordable for those reliant on housing benefits. The extent to which the framework of austerity reduces access to affordable housing for racialised minorities has received very little political or academic attention. At the same time, increasing control of migration has excluded or significantly limited access to housing based on the real or perceived citizenship status of racialised minorities (Lukes et al. 2019).

Housing policy has barely engaged with persistent racial inequalities in housing experiences in the last sixty years. The *Race Relations Act 1976* did include protections against racial discrimination in housing, and there was some funding to enable black-led housing associations to be established in the 1980s. At the local level, there have also been initiatives from local councils and, more recently, housing providers, but there is no evidence of a sustained approach to address the persistent housing inequalities experienced by racialised minorities.

## 7. Discussion and Conclusions

Housing studies tends to be conceptualised as a place-based discipline that fails to pay adequate attention to the characteristics, experiences, and aspirations of those who live in housing (Slater 2021; Allen 2009). Critical research engages with financialisation, gentrification, asylum seeker accommodation, housing activism, and the Grenfell disaster (Rolnik 2015; Lees et al. 2010; Darling 2022; Gray 2018; Apps 2022). These provide a framework for reconceptualising housing studies with the lived experience of residents central to the analysis. This also implies much more critical engagement with housing policies based on evaluation of the extent to which they are designed to address persistent housing inequalities. The article has demonstrated the importance of recognising the hostility generated by the racism and xenophobia of political and media discourse and

challenging the impact of these toxic narratives that provide individualistic explanations of housing inequality and engage with the legal, policy, and practice framework that perpetuates them. At a local level, the provision of safe, secure, adequate, and affordable housing is often not the priority for those engaged professionally with housing. In such a contested space, researchers need to consider the impact of the work they do and align their approach with the interests of those racialised and classed groups who experience housing inequalities.

Housing laws, policies, and practices in England have continued to be shaped by hostility to immigrants and racial minorities, and the consequences are persistent housing inequalities (Bramley et al. 2022; Lukes et al. 2019). These inequalities are exacerbated by the lack of control over the private rented sector and the financial dominance in the governance and regulation of social housing. There is a case for changes to national policy, and there is some attraction to focusing on this area because of the recent change in government. Developing national policies on housing and welfare that address racial inequality are also likely to address the poorer outcomes experienced by those disadvantaged by existing practices. Critical housing research and campaigning groups have used their evidence to attempt to influence national policy (Shelter 2024). The best that this can achieve would be to provide a framework enabling effective action to take place at a local level. The way that racial equity in housing will be achieved is through these local actions, and this is the area in which housing research can contribute to the evidence base that can be used by local organisations, tenants' unions, and others to challenge the way that this framework is interpreted and implemented. The devolution of housing and planning to local areas means that there is probably more benefit in working with areas that have already made some commitment to addressing the persistent housing inequalities experienced by racialised minorities. Within the context of devolution, change is most likely to happen through partnerships with metro mayors covering areas like London, Greater Manchester, and the West Midlands and the local councils with the responsibility for housing and planning within them. Housing research needs to take account of the contemporary and historical demographic, political, and social context when framing their analysis of local housing realities. This includes the traditional tools of housing needs analysis and modelling of the quality, affordability, and security of the existing housing stock and planned developments, as well as exclusionary policies and practices that disadvantage racialised minorities. Engaging with local campaigns for better housing through organisations such as tenants' unions is the most likely route to change institutional behaviours in engaging with all tenants, enforcing safety standards, and planning for future needs. The focus on local campaigns provides the opportunity to engage with comparative research, both within England and internationally.

**Funding:** This research received no external funding.

**Institutional Review Board Statement:** The study was conducted in accordance with the Declaration of Helsinki, and approved by the Proportionate UREC of the University of Manchester (2022-15607-26237 on 5 December 2022).

**Informed Consent Statement:** Informed consent was obtained from all subjects involved in the study.

**Data Availability Statement:** The original data used in this study are available from the UK Data Service, the Ahmed Iqbal Ullah Race Relations Archive reference no GB3228.5/3/43-55 and the George Padmore Institute reference no NEW/14.

**Conflicts of Interest:** The author declares no conflicts of interest.

## Notes

1    For a discussion about the ways that race and racism are framed, this summary paper (https://cdn.prod.website-files.com/6148
     8f992b58e687f1108c7c/61bccd6b2a02335f9b77b3b7_Runnymede%20Reframing%20Racism%20TUC%20briefing.pdf) (Accessed
     on 4 July 2024) from the Runnymede Trust and the TUC provides useful insights into how to write/talk about it.

2    The Decent Homes Standard was developed by the government to establish what needed to be done with the investment in
     improvements to social housing. These included meeting minimum standards in terms of hazards, being in a reasonable state of
     repair, reasonably modern facilities and services, and a reasonable degree of thermal comfort.

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
