# Peer review of "Structural Disadvantage in Housing Opportunities"

_socsci, doi:10.3390/socsci13090460_

Round 1

Reviewer 1 Report

Comments and Suggestions for Authors

The article deals with a very topical and relevant issue from the point of view of the Social Sciences in contemporary societies, in this case in England, which consists of the issue of structural and institutional racism that is reflected across the various dimensions of the tip in racialised forms of social inequality, namely the racialisation of housing.  

However, the article is essentially a reflection on housing policies in England and their racialising effects. It is therefore a purely descriptive text in which there is no empirical research, i.e. there are no research results, nor the involvement of participants who could provide clues to understanding and explaining the phenomenon.

Thus, the following aspects stand out:

1) In terms of theory and contextualisation - it presents a good framework for housing policies and discourses on race and racialisation. The use of the concept of race, applied to human beings, is controversial and refers to biological characteristics that were deconstructed by life sciences some decades ago. What exist are different cultures and cultural practices and ways of life, and this approach is not portrayed here. 

For example, it presents figure 1, taken from Finney et al., 2024, which refers to the experiences of racism lived by different ethnic minorities when looking for a home, but does not explore these differentiations, with those who have it easier being white British or not and those who have it harder being black and Gyspy, Travellers or Roma. Such a revealing panorama certainly deserved a more in-depth analysis. 

It also seems to me that the article would gain greater breadth if it extended the field of analysis to other countries, particularly other countries in the United Kingdom. While this is undoubtedly an excellent retrospective and scenario of housing policies, it is nonetheless a limited subject. 

Another topic absent from the text is the framing of migratory reception and the origins of those looking for housing and struggling with the racialisation of housing policies - for example, what about immigrants or refugees from non-European countries and those from other European countries? Are there differentiations? In particular, what about immigrants from the Commonwealth? From a sociological point of view, it would be relevant to delve into these theoretical issues in order to help understand the differences and dehomogenise the perspective of analysis.

Another topic that could enrich the text is research into urban sociology (based on the impact of the Chicago School right up to contemporary sociology) and the "marks of distinction" (in the sense of distinction as worked out by Pierre Bourdieu, 1979), social, social class and spatial distinctions, which translate into different types of housing and ways of living. 

The lack of empirical research impoverishes the article. How do racialised people live their daily lives in this racial corset, in a deeply racialised society?

2) From the point of view of the structure and style of the writing, a number of aspects are also noteworthy:

- The author refers several times to expressions such as "I believe", "my position" or "my "worldview" to give an account of his point of view on the subject he is analysing. However, in the social sciences, the application of the scientific method requires a break with common sense and the personal perspectives of researchers. Therefore, the text must be revised in this respect: the personal perspectives of the authors of the text do not matter here. At least in this field of scientific dissemination. 

- The article does not have a methodology section, which makes it difficult to understand the collection of documents and the systematisation of information and bibliography.

- In terms of the organisation of the sections, the article would benefit in terms of clarity from a more classic structure: introduction, contextualisation and theoretical framework, methodology, presentation of results, discussion and conclusion.

- The discussion section is minimalist. It should be developed and cross-referenced with the main theories on racism and racialised minorities, as well as the impact of structural inequality on their lives. 

Author Response

The article deals with a very topical and relevant issue from the point of view of the Social Sciences in contemporary societies, in this case in England, which consists of the issue of structural and institutional racism that is reflected across the various dimensions of the tip in racialised forms of social inequality, namely the racialisation of housing.  

However, the article is essentially a reflection on housing policies in England and their racialising effects. It is therefore a purely descriptive text in which there is no empirical research, i.e. there are no research results, nor the involvement of participants who could provide clues to understanding and explaining the phenomenon.

Thus, the following aspects stand out:

1) In terms of theory and contextualisation - it presents a good framework for housing policies and discourses on race and racialisation. The use of the concept of race, applied to human beings, is controversial and refers to biological characteristics that were deconstructed by life sciences some decades ago. What exist are different cultures and cultural practices and ways of life, and this approach is not portrayed here. 

For example, it presents figure 1, taken from Finney et al., 2024, which refers to the experiences of racism lived by different ethnic minorities when looking for a home, but does not explore these differentiations, with those who have it easier being white British or not and those who have it harder being black and Gyspy, Travellers or Roma. Such a revealing panorama certainly deserved a more in-depth analysis. 

It also seems to me that the article would gain greater breadth if it extended the field of analysis to other countries, particularly other countries in the United Kingdom. While this is undoubtedly an excellent retrospective and scenario of housing policies, it is nonetheless a limited subject. 

Another topic absent from the text is the framing of migratory reception and the origins of those looking for housing and struggling with the racialisation of housing policies - for example, what about immigrants or refugees from non-European countries and those from other European countries? Are there differentiations? In particular, what about immigrants from the Commonwealth? From a sociological point of view, it would be relevant to delve into these theoretical issues in order to help understand the differences and dehomogenise the perspective of analysis.

Another topic that could enrich the text is research into urban sociology (based on the impact of the Chicago School right up to contemporary sociology) and the "marks of distinction" (in the sense of distinction as worked out by Pierre Bourdieu, 1979), social, social class and spatial distinctions, which translate into different types of housing and ways of living. 

The lack of empirical research impoverishes the article. How do racialised people live their daily lives in this racial corset, in a deeply racialised society?

2) From the point of view of the structure and style of the writing, a number of aspects are also noteworthy:

- The author refers several times to expressions such as "I believe", "my position" or "my "worldview" to give an account of his point of view on the subject he is analysing. However, in the social sciences, the application of the scientific method requires a break with common sense and the personal perspectives of researchers. Therefore, the text must be revised in this respect: the personal perspectives of the authors of the text do not matter here. At least in this field of scientific dissemination. 

- The article does not have a methodology section, which makes it difficult to understand the collection of documents and the systematisation of information and bibliography.

- In terms of the organisation of the sections, the article would benefit in terms of clarity from a more classic structure: introduction, contextualisation and theoretical framework, methodology, presentation of results, discussion and conclusion.

The discussion section is minimalist. It should be developed and cross-referenced with the main theories on racism and racialised minorities, as well as the impact of structural inequality on their lives

Responses

The article has been developed from a keynote speech to the Housing Studies Association and is intended to identify gaps and issues in the scholarly research into housing with respect to race (action 1).

There is no consensus on the concept of race and in this article, I try to show how policy discourses have drawn on migration status to racialise minorities in different ways and at different times.  

Agreed (action 2)

The rationale is that racism and attitudes are a particularly English outlook with quite different attitudes in other UK countries. There are also significant differences between different areas within England. Comparative research could help to explain this but is beyond the scope of this article. The potential is highlighted in the discussion and conclusion (Action 3).

Accepted to some extent but the main point is the commonality between the experiences of migrants and racialised minorities regardless of origin.  This leads to the importance of context in explaining the shifting nature of structural discrimination in housing opportunities as outlined throughout the article (Action 4).

Whilst this may deserve consideration in future research the Chicago school specifically desccribed black Americans as a different case and the work on housing based on Bourdieu’s theoretical framework (e.g. Loic Wacquant) largely ignores race as a structural factor. This may in part be because of the French failure to acknowledge race.

The article calls for housing researchers to carry out empirical research with an understanding of the need to incorporate the lived experience of racialised minorities rather than attempting to cover this.  For an example see the report by David Robinson et al., (2022) Race Equality in Housing: A Review of the Policy Approach in England, Scotland and Wales. It also reframes the evidence as parts of empirical research carried out in other contexts (Action 5).

Positionality in regard to concepts where there is a lack of consensus in the research field is critical to knowledge.  My personal beliefs shape the knowledge I generate in my work and failure to acknowledge that is a weakness in this field. I have rephrased the statements to provide a less individualistic account.

Agreed to some extent – proviso that the aim is not to incorporate significant empirical research Action 6.

Agreed to some extent.  The expectation that main theories on racism and racialised minorities and the impact of structural inequality on their lives will be considered is unrealistic in the scope of this article.

Actions

Action 1: Updated paragraph 1 to reflect the purpose of the article, restructured article and identified three sections as based on empirical research

Action 2: Extended discussion of figure 1 to highlight contrast between experiences of racism when seeking housing

Action 3: Highlighted the potential for comparative research to illuminate particular aspects of racism within different cases in the discussion and conclusion section.

Action 4: Emphasise the importance of understanding the context of race and racialisation in different places.

Action 5: Incorporate a brief discussion of the methods used and restructure the evidence provided to demonstrate its empirical nature.

Action 6: Revised structure and enhance the discussion section to draw together an agenda for housing research that critically engages with race and racialisation.

Reviewer 2 Report

Comments and Suggestions for Authors

Dear Authors,

Thankyou for the opportunity to review your paper.  The topic of racism within the development of housing policy and practice is certainly an area requiring more scholarly attention.  I have a series of comments and suggestions for your consideration - 

- P1. Abstract and Introduction.  Your paper makes a number of claims about taking a "critical approach" to exploring the development of housing policy.  A sentence about what this means and in your abstract would be useful as well a few sentences in your introduction about what this means and why you chose this approach.  

- Your essay has a critically reflective nature and I believe you shouldn't be afraid to refer to this.  As such the actual methodology could be outlined more clearly and succinctly.

- P2. Discussion about Racism. The paragraph on definition of racism needs some work and a contextualised discussion of the definition of racism. There also needs to be more intext referencing in this paragraph.

- Given the paper claims to take a "critical perspective" I did wonder the authors did not also take the opportunity to discuss racism and "privilege" in housing policy and housing systems?

- P4. First paragraph needs some editorial work.  There are two sentences that begin the same way in succession.  "This approach...." followed by a third sentence "This shifted....".  Try to vary your writing style where you can.

- Intext reference needed to support your statement in last paragraph about "Moss side in Manchester'.

- Consistency in writing legislation. Typically legislation is  italicised ie. Aliens Act 1905 

- P6. Intext reference needed to support the discussion on "Decent Home Standard" as this is a new term in the paper.

- P7. Intext reference to 2021 Census needed.

- p8. The table on experiences of racism in housing could be used to much greater effect if the table was introduced much earlier in the paper. I would suggest moving the table to section 3 on Race and migration.

Author Response

Dear Authors,

Thankyou for the opportunity to review your paper.  The topic of racism within the development of housing policy and practice is certainly an area requiring more scholarly attention.  I have a series of comments and suggestions for your consideration - 

- P1. Abstract and Introduction.  Your paper makes a number of claims about taking a "critical approach" to exploring the development of housing policy.  A sentence about what this means and in your abstract would be useful as well a few sentences in your introduction about what this means and why you chose this approach.  

- Your essay has a critically reflective nature and I believe you shouldn't be afraid to refer to this.  As such the actual methodology could be outlined more clearly and succinctly.

- P2. Discussion about Racism. The paragraph on definition of racism needs some work and a contextualised discussion of the definition of racism. There also needs to be more intext referencing in this paragraph.

- Given the paper claims to take a "critical perspective" I did wonder the authors did not also take the opportunity to discuss racism and "privilege" in housing policy and housing systems?

- P4. First paragraph needs some editorial work.  There are two sentences that begin the same way in succession.  "This approach...." followed by a third sentence "This shifted....".  Try to vary your writing style where you can.

- Intext reference needed to support your statement in last paragraph about "Moss side in Manchester'.

- Consistency in writing legislation. Typically legislation is  italicised ie. Aliens Act 1905 

- P6. Intext reference needed to support the discussion on "Decent Home Standard" as this is a new term in the paper.

- P7. Intext reference to 2021 Census needed.

p8. The table on experiences of racism in housing could be used to much greater effect if the table was introduced much earlier in the paper. I would suggest moving the table to section 3 on Race and migration

Responses

The reviewer comments were helpful and have addressed them

Actions

Action 7: Update abstract and introduction to reflect the critical approach

Action 8: Outline methodology succinctly

Action 9: Developed definition of racism and used external references

Action 10: Incorporated privilege in section on race and migration

Action 11: Revised writing style substantially through the paper

Action 12: Added references to paragraph starting Migrants to Moss Side

Action 13: Italicised references to legislation

Action 14: Footnote added to define Decent Homes Standard

Action 15: Added intext references to census data together with links to recreate the tables used for this analysis

Action 16: Figure 1 moved to earlier section and results discussed

Round 2

Reviewer 1 Report

Comments and Suggestions for Authors

The article has undergone a significant improvement with the new theoretical inputs and restructuring of the sections, making it clearer and more explicit. 

Although I continue to argue that an international comparison would be very useful, namely with countries that have different housing policies and are also major recipients of immigrants, in order to broaden the reflexivity on housing policies in the UK, I noted that the main suggestions made in the previous review have been included.  I also continue to argue that racist practices are transversal in contemporary societies, where access to housing is only one of its visible dimensions, and that social analysis must consider its multiple dimensions.

The suggestion of theoretical support from the Chicago School and some authors such as Pierre Bourdieu and Loic Wacquant had nothing to do with the concept of racism based on skin color but with the "neighborhood effect", in which the accommodation of certain families migrants and non-migrants in deprived and marginalized spaces of the urban fabric triggers ghettoization phenomena, which mark successive generations.

Reviewer 2 Report

Comments and Suggestions for Authors

Thankyou for addressing my earlier comments and feedback.  The paper has been revised and is now at an academic standard ready for publication.